# Uncovering Novel Roles of miR-122 in the Pathophysiology of the Liver: Potential Interaction with NRF1 and E2F4 Signaling

**DOI:** 10.3390/cancers15164129

**Published:** 2023-08-16

**Authors:** Martha Paluschinski, Jessica Schira-Heinen, Rossella Pellegrino, Lara R. Heij, Jan Bednarsch, Ulf P. Neumann, Thomas Longerich, Kai Stuehler, Tom Luedde, Mirco Castoldi

**Affiliations:** 1Department of Gastroenterology, Hepatology and Infectious Diseases, Medical Faculty and University Hospital, Heinrich Heine University Dusseldorf, 40225 Dusseldorf, Germany; martha.paluschinski@hhu.de (M.P.); tom.luedde@med.uni-duesseldorf.de (T.L.); 2Department of Neurology, Medical Faculty, Heinrich-Heine-University, 40225 Dusseldorf, Germany; j.schira@hhu.de; 3Molecular Proteomics Laboratory (MPL), Institute for Molecular Medicine, Heinrich-Heine-University, 40225 Dusseldorf, Germany; kai.stuehler@hhu.de; 4Institute of Pathology, University Hospital Heidelberg, 69120 Heidelberg, Germany; rossella.pellegrino@med.uni-heidelberg.de (R.P.); thomas.longerich@med.uni-heidelberg.de (T.L.); 5Department of Surgery and Transplantation, University Hospital RWTH Aachen, 52074 Aachen, Germany; lheij@ukaachen.de (L.R.H.); jbednarsch@ukaachen.de (J.B.); uneumann@ukaachen.de (U.P.N.)

**Keywords:** miR-122, polyribosomes, liver cancer, NRF1, E2F4, G6PD

## Abstract

**Simple Summary:**

The microRNA miR-122 plays a crucial role in liver function, but its full impact on gene regulation and disease mechanisms remains unclear. To better understand miR-122, two different methods were used: studying its effects on translation through ribosome occupancy analysis and at the protein level. We found that miR-122 could potentially antagonize the activity of disease-related transcription factors and influence the expression of malignancy-associated proteins. Many of these proteins have been validated as novel targets of miR-122 and linked to liver disease biogenesis. In summary, this study sheds new light on the role of miR-122 by proposing a novel molecular mechanism on how disruption of this miRNA may contribute to the development of liver disease.

**Abstract:**

MicroRNA miR-122 plays a pivotal role in liver function. Despite numerous studies investigating this miRNA, the global network of genes regulated by miR-122 and its contribution to the underlying pathophysiological mechanisms remain largely unknown. To gain a deeper understanding of miR-122 activity, we employed two complementary approaches. Firstly, through transcriptome analysis of polyribosome-bound RNAs, we discovered that miR-122 exhibits potential antagonistic effects on specific transcription factors known to be dysregulated in liver disease, including nuclear respiratory factor-1 (NRF1) and the E2F transcription factor 4 (E2F4). Secondly, through proteome analysis of hepatoma cells transfected with either miR-122 mimic or antagomir, we discovered changes in several proteins associated with increased malignancy. Interestingly, many of these proteins were reported to be transcriptionally regulated by NRF1 and E2F4, six of which we validated as miR-122 targets. Among these, a negative correlation was observed between miR-122 and glucose-6-phosphate dehydrogenase levels in the livers of patients with hepatitis B virus-associated hepatocellular carcinoma. This study provides novel insights into potential alterations of molecular pathway occurring at the early stages of liver disease, driven by the dysregulation of miR-122 and its associated genes.

## 1. Introduction

The liver plays a critical role in maintaining metabolic homeostasis by synthesizing, storing, and redistributing metabolites. It also acts as the body’s primary organ for detoxification, making it susceptible to toxicity. Acute or repeated insults to the liver are known risk factors for liver disease [1]. MicroRNAs (miRNAs) are small noncoding RNAs that regulate gene expression at the post-transcriptional level. Extensive evidence suggests that miRNAs have the ability to control networks comprising hundreds of genes, thus triggering pleiotropic effects in response to physiological and pathological changes in their expression. Among these miRNAs, miR-122 has been extensively studied and implicated in various aspects of hepatic metabolism, such as cholesterol metabolism [2] and iron homeostasis [3]. Additionally, miR-122 was shown to possess tumor suppressor and anti-inflammatory properties [4]. Notably, miR-122 was found significantly downregulated in the livers of approximately 70% of hepatocellular carcinoma (HCC) patients [5], and lower levels of miR-122 have been associated with shorter recurrence times in resected HCC patients [6]. Studies on miR-122 knockout (KO) animals have revealed that while these animals are viable, their livers exhibit infiltration by inflammatory cells, leading to the development of steatohepatitis, fibrosis, and HCC [4], while miR-122 inhibition in human liver organoids has been shown to lead to inflammation [7]. Collectively, these findings support the existence of a link among miR-122, inflammation, and liver cancer. Despite the extensive knowledge on miR-122, numerous questions remain unanswered. For instance, the mechanisms underlying miR-122 downregulation during the onset of liver diseases are still poorly understood. Is the downregulation of miR-122 merely a consequence of healthy parenchyma loss, or is it a physiological process with a specific regulatory function that has gone awry? Furthermore, what are the molecular networks that underlie its functional role? To further explore the impact of miR-122 on liver pathophysiology, a combination of transcriptomic and proteomic approaches was applied, providing evidence that alterations in miR-122 levels have the potential to modify the translational activity of thousands of genes. Our data suggest that miR-122 potentially modulate genes downstream of several transcription factors (TFs), including nuclear respiratory factor-1 (NRF1), transcription factor E2F4 (E2F4), and Yin Yang 1 (YY1). Through our investigation, several novel miR-122 targets were identified and validated. Notably, elevated expression of these genes has been linked to an unfavorable clinical outcome in HCC patients, suggesting their potential prognostic value. Overall, this study sheds new light on the potential changes in signaling pathways that may occur during the onset of liver disease, which may be potentially linked to alterations in miR-122 and its associated mRNA targets.

## 2. Results

### 2.1. Preparation and Isolation of Polyribosomes from Sucrose Gradient

Despite the extensive literature on miRNA biology, the identification of miRNA targets remains a contentious issue. It was initially proposed that miRNAs inhibited protein synthesis via inducing degradation of their mRNA targets [8,9]. On the contrary, studies on isolated polyribosomes suggested that miRNAs primarily cause mRNA destabilization [10,11]. Consequently, analyzing polyribosomes could facilitate the identification of bona fide miRNA targets and their associated regulatory network (reviewed in [12]). To evaluate the efficacy of polyribosome analysis in identifying genes responsive to miR-122, human hepatoma cell lines (Huh-7) were transfected with miR-122 mimics or inhibitors (antagomiRs). Polyribosomes were then isolated through fractionation on sucrose gradients. As a proof of principle, the relative distribution of genes of interest (GOI) on the polyribosomes was assessed using quantitative real-time PCR (Figure 1). We found that miR-122 overexpression led to a significant increase in the levels of miR-122 in the polyribosomal fractions (Figure 1B). In contrast, the expression and distribution of other miRNAs, here illustrated by miR-192, remained unaffected (Figure 1C). Consistently, the analysis of the best-known direct target of miR-122, SLC7A1 (high-affinity cationic amino-acid transporter 1 [13]), identified a significant increase in the abundance of SLC7A1 RNA in the polyribosomal fractions isolated from antagomir-transfected cells (Figure 1D). Notably, the distribution on polyribosimal fractions of hemochromatosis (HFE), hemojuvelin (HJV), and hepcidin (HAMP) transcripts, all genes that were previously reported to respond to miR-122 [3], was found to be affected by the modulation of miR-122 levels (Appendix A). Conversely, the distribution of genes that are not expected to respond to miR-122, including transferrin receptor 2 (TFR2), bone morphogenetic protein 6 (BMP6), L-ferritin, and growth differentiation factor 15 (GDF15; Figure 1E and Appendix A), were not significantly affected. Thus, these results support the conclusion that known miR-122-responsive genes can be identified by analyzing the distribution of genes of interest (GOIs) on polyribosomes.

### 2.2. Modulation of miR-122 Expression Results in Changes in Polyribosome Occupancy for Thousands of Transcripts

To better comprehend the impact of miR-122 on the cellular transcriptome and its molecular networks, we took a comprehensive approach. RNAs were isolated from the individual polyribosomal fractions of cells transfected with either miR-122 mimics (122-MIM) or antagomiRs (122-PM). These RNAs were then combined to create polyribosomal pools of decreasing densities [i.e., P1 = heavy (fractions A1–A4), P2 = middle (fractions A5–A8), and P3 = light (fractions A9–A12); visualized in Figure 2A]. To conduct an in-depth analysis, these six different RNA pools (i.e., three densities for two conditions for two different experiments) were hybridized to microarrays, and bioinformatics analysis was carried out to identify miR-122-responsive genes. Specifically, AltAnalyze [15] was used to compare the changes in the relative amount of genes across the three different pools (i.e., P1 vs. P2, P2 vs. P3, and P1 vs. P3) within each condition (i.e., 122-MIM or 122-PM). Moreover, comparisons were carried out across the conditions (i.e., 122-MIM vs. 122-PM) within each pool (i.e., P1, P2 or P3) for a total of nine comparisons. In detail, we measured (i) the significant shift of genes from high-density to low-density fractions in cells transfected with the miR-122 mimic (122-MIM; Figure 2B,C), which led to the shift of 6621 transcripts toward lighter-density polyribosomes indicating a reduced translational turnover of the associated genes in response to miR-122 upregulation; (ii) the significant shift of genes from low-density to high-density fractions in cells transfected with the antagomiR (122-PM; Appendix A), which identified the shifting of 6163 transcripts toward polyribosomes with heavier density, indicating an increase in translational turnover of these genes in response to miR-122 inhibition; and (iii) the significant change in the expression of genes across the same fractions in 122-MIM versus 122-PM (Appendix A), which identified the significant enrichment of 1028 transcripts in polyribosomes isolated from 122-PM-transfected cells.

### 2.3. Potential Regulatory Role of miR-122 on NRF1-, E2F4-, and YY1-Mediated Gene Regulation

To explore the functional significance associated with the shift in mRNA occupancy in response to changes in miR-122, the GO-Elite algorithm [16] was used to analyze the differentially regulated genes (DEGs) from transcriptomic data. GO-Elite allows the identification of common biological ontology pathways that describe a specific set of genes. Specifically, GO-Elite identified a significant enrichment for genes associated to the transcription factors (TFs) YY1, E2F4, NRF1, and Forkhead box P3 (FOXP3; Figure 2D and Appendix A). Markedly, these transcription factors have been found to be dysregulated in patients with liver diseases including steatosis, nonalcoholic steatosis, and HCC [17,18,19,20]. Interestingly, the mining of sequencing data of a cohort of patients with liver hepatocellular carcinoma (LIHC) included in The Cancer Genome Atlas (TCGA) shows that YY1, E2F4, and NRF1 expressions were significantly upregulated in the liver of HCC patients (Appendix A). Moreover, the Kaplan–Meier curves from these data show that LIHC patients with lower YY1, E2F4, and NRF1 expression have a significantly higher probability of survival (Appendix A). Contrariwise, FOXP3 (Figure 2D), which is mainly expressed in regulatory T-cells (Treg [18]), was found to be significantly downregulated in TCGA LIHC cohort, and LIHC patients with higher FOXP3 expression had a significantly higher probability of survival (Appendix A). The upregulation of YY1, E2F4 and NRF1 in the liver of HCC patients was also confirmed through the analysis of cohorts of patients downloaded from GEO (i.e., GSE62232 and GSE6764; Appendix A). Significantly, the application of GO-Elite to the analysis of patients’ cohorts with grade III HCC (accession number GSE45050; [21]) identified a significant enrichment for genes associated to YY1, NRF1, and FOXP3 transcription factors (Appendix A), indicating that modulations of similar changes might be extrapolated through the analysis of the whole liver. Remarkably, a similar enrichment for YY1, NRF1, E2F4, and FOXP3 TFs was also identified by GO-Elite in both the livers of miR-122 KO animals (GSE31453) and in the mouse model for liver cancer (GSE27713; Appendix A).

### 2.4. miR-122-Responsive Genes are under NRF1, E2F4, and YY1 Transcriptional Control

To validate the microarray and GO-Elite data, we followed a rigorous selection process resulting in the selection of 43 genes based on specific criteria. Firstly, a sub-list of genes was generated by selecting TF-associated genes [Appendix A, lists for individual TFs are found in Appendix A] responsive to miR-122 and present in at least two polyribosomal pool comparisons. Secondly, to identify potential miR-122 targets, this sub-list was overlaid with a list of predicted miR-122 targets generated by *miRWalk* [22]. Subsequently, genes that exhibited less than a 1.5-fold differential expression were filtered out. Finally, the remaining genes were thoroughly reviewed in the literature, specifically focusing on their links to “liver disease”, “infection”, “inflammation”, or “cancer”, generating a final list of 45 genes (Appendix A). Next, qPCR was used to assess the expression of the target genes in Huh-7 cells following transfection with 122-MIM (Figure 3), resulting in a significant reduction in the expression of 24 genes (i.e., ~56% of the selected genes). Specifically, the genes affected after 24 h were BAG1, BAX, CCDC43, CD47, DIXDC1, DSG2, F2RL2, G3BP2, KPNA6, KPNB1, NUP210, P4HA1, PCDC4, PDCD2, and TBC1D22B. After 48 h, the genes affected were CMTM7, HCFC1, KIF1B, KIF3A, MINK1, PDCD2, RNF26, SPRED2, and TNPO1 (Figure 3B,C). It is important to note that the identification of “unresponsive” genes does not exclude them from potentially being targeted by miR-122, as their regulation could occur at the translational level. Furthermore, we made an intriguing observation that the overexpression of miR-122 led to a significant reduction in E2F4 and NRF1 levels, while the expression of YY1 remained unaffected (Figure 3B). Notably, GO-Elite enrichment analysis also identified a significant enrichment of genes associated with the FOXP3 transcription factor (Figure 2D, Appendix A). Although FOXP3 mRNA was not detected in Huh-7 cells, our analysis suggested that Huh-7 cells expressed a subset of FOXP3-responsive genes that also responded to changes in miR-122 levels (Appendix A). Overall, these findings highlight that alterations in miR-122 levels can influence the ribosomal occupancy of numerous transcripts, which have been linked to chronic liver diseases and liver cancer. Consequently, we propose that miR-122 may directly or indirectly contribute to modifying the translational turnover of gene subsets linked to the transcription factors E2F4, NRF1, and YY1, as well as their associated molecular networks.

### 2.5. Proteomic Analysis Uncovers Proteins Linked to Liver Diseases and Reveals Potential Role of miR-122 in Regulating Energy Metabolism and EV Secretion

Further investigations were carried out to evaluate the impact of changes in miR-122 expression on protein levels. Huh-7 cells were transfected with either miR-122 mimics (122-MIM) or antagomiRs (122-PM). The proteins extracted from these cells were then analyzed by mass spectrometry (Figure 4). Among the 1940 identified proteins, 504 proteins exhibited significant regulation (*p* ≤ 0.05; Figure 4A,B and Appendix A). To gain insight into the functional implications of these changes, ShinyGO [24] was used to perform enrichment analysis. The results, shown in Figure 4C, shed some light onto the potential effects of miR-122 alterations on the cellular proteome. Notably, an examination of the GO terms related to “biological processes” suggested that changes in miR-122 expression might affect energy metabolism (Figure 4C, top panel). Additionally, an analysis of the GO terms related to “cellular components” identified a significant reduction in terms associated with “extracellular vesicles”, while a significant enrichment of terms associated with mitochondria and organelle was detected (Figure 4C, lower panel). These findings support existing evidence suggesting the involvement of miR-122 in regulating both mitochondrial [25] and energy metabolism [26].

### 2.6. miR-122 Regulates the Expression of Proteins Potentially Contributing to Liver Cancer Development

To validate the proteomic data, a rigorous selection process was followed. First, a list of proteins that showed significant responsiveness to changes in miR-122 levels was compiled. From this list, we selected 43 proteins that exhibited more than 1.5-fold change and were predicted as potential miR-122 targets by the *miRWalk* target prediction database. To examine the connections between transcriptomic and proteomic data, transcription factor enrichment analysis (TFEA) was performed. For this purpose, ShinyGO was used to cross-reference the list of 43 miR-122-responsive proteins with the ENCODE-motifs database [28]. The TFEA results revealed that approximately 90% (40 out of 43) of the genes contained binding motifs for E2F4, and around 86% (37 out of 43) contained binding motifs for NRF1 (Appendix A). Next, a literature search was conducted to focus on proteins specifically associated with liver disease in general, which selected 20 proteins (Appendix A and Appendix A), seven of which (i.e., CEP55, CLIC1, EPS15L1, G6PD, KIF11, SLC1A5, and TK1) were found to be specifically associated with hepatocellular carcinoma. This conclusion was supported not only by literature mining but also through the analysis of publicly available databases, including GEO (i.e., accession numbers GSE62232 and GSE6764, Appendix A) and TCGA (Appendix A). To this end, detailed analysis of the sequencing data from TCGA LIHC cohort showed that CEP55, CLIC1, EPS15L1, G6PD, KIF11, SLC1A5, and TK1 were significantly upregulated in the livers of HCC patients. Notably, the Kaplan–Meier curve indicated that patients with lower expression of these genes had a significantly higher probability of survival. Next, the regulatory effect of miR-122 on the mRNAs encoding these proteins was evaluated. Here, we show that a significant decrease in the mRNA levels of these genes was measured in 122-MIM-transfected cells (Figure 4D), supporting the conclusion that miR-122 affects the stability of these transcripts, either directly or indirectly.

### 2.7. CEP55, CLIC1, G6PD, KIF11, SLC1A5, and TK1 but not EPS15L1 are Direct Targets of miR-122

To investigate the influence of miR-122 on the stability of specific transcripts, the full-length 3′UTRs of CEP55, CLIC1, EPS15L1, G6PD, KIF11, SLC1A5, and TK1 were cloned in the pMIR(+) and pMIR(−) Luciferase reporter vectors, and co-transfected with miR-122 mimics into HEK293 cells (Figure 5). Here, we show that, in cells transfected with miR-122 mimics, the Luciferase activity of the pMIR(+) vectors (i.e., carrying the 3′UTRs in their native orientation) was significantly reduced. This effect was observed for all 3′UTRs, except for EPS15L1. Conversely, miR-122 overexpression had no impact on the Luciferase activity of the pMIR(−) control vectors, which contained the 3′UTRs cloned in the reverse complement orientation. Consequently, it can be inferred that miR-122 directly regulates the expression of CEP55, CLIC1, G6PD, KIF11, SLC1A5, and TK1. However, in the case of EPS15L1, the regulation by miR-122 is likely indirect, as demonstrated by a noticeable decrease in EPS15L1 protein levels observed in Western blot analysis (Figure 5B) upon miR-122 upregulation.

### 2.8. The Expression of G6PD and miR-122 is Inversely Correlated in the Liver of HBV Patients with HCC

Our data indicate that miR-122 directly controls the translation of genes that may contribute to HCC development in human (Figure 5A). In order to delve deeper into these findings and establish a connection to human hepatocarcinogenesis, the focus was placed on G6PD. Consistent with previous research [29], our study confirms that miR-122 interacts with G6PD 3′UTR to regulates its expression both at the mRNA (Figure 4D) and at the protein level (Figure 5C). 

Furthermore, G6PD protein levels were measured by Western blot in the liver tissue of HCC patients and found to be significantly upregulated (Figure 6, patient description can be found in the Appendix A). Notably, this data is in agreement with the analysis of G6PD expression in the TCGA cohorts of LIHC, which indicates that G6PD might serve as a prognostic marker for HCC, with higher expression associated with a worse prognosis (Figure 7A). Remarkably, analysis of miR-122 expression in TCGA LIHC cohorts indicated the inverse correlation between G6PD and miR-122 in liver cancer patients (Appendix A). Multiple studies have reported an elevation in G6PD levels in various types of human cancer, including HCC [30,31,32].

In order to further investigate the link between miR-122, G6PD in liver cancer development, we specifically focused on HCC patients with or without hepatitis B virus (HBV) infection. Independent studies have shown that, while HBV infection enhances G6PD expression through hepatitis B viral protein X (HBx)-mediated Nrf2 activation [33], miR-122 was downregulated in the livers of HBV-infected patients [34]. To assess the relationship between miR-122 and G6PD in the livers of HBV-infected patients, RNA samples were extracted from FFPE tissue slices of patients with HCC with HBV (*n* = 7) or without HBV infection (*n* = 21; patient description can be found in Appendix A) and analyzed by qPCR. Although no differences were observed in the overall expression of miR-122 between the two groups (Figure 7B), a trend toward increased G6PD expression (*p* = 0.0724) was observed in the tumor tissue of HBV-infected patients. Importantly, when linear regression analysis was applied to the datasets, an inverse correlation between miR-122 and G6PD was observed in the HBV-infected HCC group (R^2^ = 0.6095, *p* = 0.0383; Figure 7C), but not in the non-viral HCC group. On the basis of these findings, we propose that the HBV-mediated suppression of miR-122 may directly contribute to the upregulation of G6PD observed in the livers of patients with chronic HBV infection.

## 3. Discussion

The dysregulation of miR-122 has been implicated in the development and progression of liver diseases, although the exact link between aberrant miR-122 expression and liver disease pathogenesis remains unclear [35,36]. In our study, we aimed to investigate the functional role of miR-122 by examining changes in polyribosome-associated mRNA upon miR-122 overexpression and inhibition. Our results revealed that changes in miR-122 expression significantly affected the occupancy of multiple transcripts on polyribosomes, confirming the broad regulatory influence of miR-122. Interestingly, bioinformatics analysis identified the potential interplay between E2F4, NRF1, YY1, and FOXP3 transcription factors and a large number of miR-122-responsive genes. Notably, previous studies have linked NRF1 inactivation to the development of nonalcoholic steatohepatitis [33], while YY1 upregulation has been associated with fatty liver diseases [17], whereas E2F4 was found to promote HCC proliferation [37,38]. Furthermore, the interplay between NRF1 and E2F4 transcription factors was previously shown to contribute to cancer development and progression [39]. On the basis of these findings, we propose that miR-122 may “function” to “counteract” or “fine-tune” the activity of E2F4, NRF1, and YY1, thus contributing to the maintenance of liver homeostasis as proposed in Figure 8A. Conversely, the downregulation of miR-122 observed in patients with liver diseases, along with the consequent dysregulation of these transcription factors and their associate networks, may contribute to the “chronicization” process associated with liver diseases in these patients, as proposed in Figure 8B. Furthermore, in a recent publication, our research unveiled a significant connection among the bone morphogenetic protein 6 (BMP6), transforming growth factor beta (TGFβ), and regulation of miR-122 transcription [40], shedding light on a potential link between reduced miR-122 levels and hepatic inflammation. This pivotal finding, combined with the insights described in this manuscript, strengthens the hypothesis that the reduction in miR-122 expression is a physiological response triggered (also) by inflammation, leading to the fine-tuning of the signaling cascades downstream of E2F4, YY1, and NRF1.

Interestingly, although FOXP3 was not detectable in Huh-7 cells, our data suggest that a subset of FOXP3-associated genes is expressed in these cells and responds to miR-122. FOXP3 is primarily expressed in Treg cells [18], where it plays a crucial role in immune tolerance by limiting immune activation of the microenvironment. Analysis of TCGA data further supports the prognostic value of FOXP3 in liver hepatocellular carcinoma (LIHC), with lower expression being unfavorable (Appendix A). We propose that the increased secretion of miR-122 observed in patients with liver diseases [44] may lead to the downregulation of FOXP3-associated targets in Treg, ths contributing to the establishment of a pro-inflammatory milieu in the liver, as reported for miR-122 KO mice [4] and human liver organoids [7]. It is noteworthy that, although there are a growing number of studies describing the role played by circulating miR-122 in the progression of human diseases, including cancer [45,46], to the best of our knowledge, our work is the first to suggest that cellular miR-122 may itself have a significant impact on the cellular secretome (see Figure 4). Altogether, the proposed activities of miR-122 on cellular secretome and on FOXP3 will be evaluated in planned follow-up studies.

Although transcriptomic data provide valuable information about the cellular state, it is crucial to acknowledge that proteins are the primary agents responsible for cellular functions. Currently, the correlation between transcriptomic and proteomic data is limited [47,48,49]. The fact that changes in gene expression do not always lead to a parallel change in protein levels can be attributed both to technical constraints related to mass spectrometry, such as the inability to detect less abundant proteins [50], and to the effect of the post-transcriptional machinery on mRNA translation and protein synthesis. Therefore, transcriptomic data were combined with mass spectrometry analysis, confirming that a number of miR-122 responsive genes enriched on polyribosomes were also influenced at the protein level by changes in miR-122. Literature mining in combination with GEO/TCGA database analysis showed that a subset of miR-122-responsive proteins, including G6PD, CEP55, KIF11, SLC1A5, EPS15L1, TK1, and CLIC1, were upregulated in liver tumor tissues of HCC patients (Appendix A), with their increased expression being unfavorable in liver cancer (Appendix A). Overall, we strongly believe that assessing the impact of these genes on liver cancer’s development and progression is a crucial next step, making further studies warranted.

Given the global distribution of hepatitis B, its association with poor treatment response, and the increased risk of HCC development, we specifically investigated the functional role of miR-122 in regulating G6PD in chronic hepatitis B (CHB) patients. Enhanced G6PD activity has been linked to the upregulation of anti-apoptotic factors such as Bcl-2 and Bcl-xL [33], which contribute to promoting cell growth and tumor development [42]. Additionally, increased G6PD activity may contribute to the accelerated turnover of the pentose phosphate pathway [51]. This pathway is crucial for cancer cell growth as it provides nucleic acids and NADPH for biosynthesis, as well as glutathione for combating oxidative stress. G6PD has been suggested as a potential therapeutic target in HCC, as inhibiting its activity could help prevent tumor progression [31]. In our study, we confirmed that G6PD was a direct target of miR-122, results that are consistent with published research [29]. These findings support the hypothesis that HBV suppression of miR-122 may contribute directly to increased G6PD levels in the livers CHB patients. We propose that this may be an additional “risk factor” contributing to the increased risk of developing hepatocellular carcinoma in individuals with CHB (Figure 8B).

Notably, miR-122 achieved a significant milestone as the first miRNA to have a targeted drug, namely, miravirsen [52]. This pioneering achievement proved highly effective in suppressing viral replication in individuals suffering from chronic hepatitis C virus (HCV) infections [53]. Nevertheless, the introduction of innovative and potent pharmaceutical agents for hepatitis C treatment [54] has considerably diminished the emphasis on therapeutic strategies involving miR-122 inhibition. The findings presented within this study, coupled with our recently published research [40], propose an intriguing potential physiological role for the reduction in miR-122 in response to inflammation. These insights hold the potential to stimulate a fresh evaluation of therapeutic opportunities associated with targeting miR-122 in CLDs, as inhibiting miR-122 could potentially emerge as a valuable approach for modulating inflammatory responses within the liver, warranting further consideration and exploration.

Overall, our research confirmed the central role of miR-122 in the regulation and fine-tuning of hepatic homeostasis. In this study, we discovered a new, hitherto uncharacterized molecular network downstream of miR-122, highlighting the possible interaction between miR-122 and the transcription factors NRF1, E2F4, and YY1. We propose that a combination of physiological and pathological signals, yet to be established, drives the downregulation of miR-122 in parenchymal cells and, possibly, its upregulation in Treg cells, thus contributing to the establishment of a pro-inflammatory and pro-tumorigenic environment in the liver.

## 4. Materials and Methods

### 4.1. Isolation and Fractionation of Polyribosomes

Polyribosomes were isolated from Huh-7 cells transfected with miR-122 mimic or miR-122 inhibitor for 48 h. Cells were treated with 200 µg/mL cycloheximide (Sigma-Aldrich, Taufkirchen, Germany, C4859-1ML) for 10 min, washed with PBS/100 µg/mL cycloheximide, and centrifuged (1000 rpm/4 °C/5 min). The pelleted cells were lysed in 750 µL of lysis buffer (5 mM Tris HCl pH 7.5, 1.5 mM KCl, 2.5 mM MgCl_2_, 0.2 mM cycloheximide) supplemented with 120 U/mL RNase Inhibitor (RiboLock, Thermo Fisher Scientific, Meerbusch, Germany, EO0381), 120 U/µL DNase I (New England BioLabs, Frankfurt a.M., Germany, M0303S), 0.5% sodium deoxycholate (Sigma-Aldrich, Taufkirchen, Germany, 30970-25G), and 0.5% Triton X-100. Cell nuclei were removed by centrifugation; then, the supernatant was layered on top of a linear sucrose gradient (10–50% sucrose) loaded in a SW40 Ti swinging-bucket rotor (Beckman Coulter, Krefeld, Germany) and centrifuged (33.500 rpm/4 °C/3 h, brake off) in an Optima XPN-80 ultracentrifuge (Beckman Coulter, Krefeld, Germany). Sucrose gradients were eluted into a BR-188 density gradient fractionation system (Brandel Inc, Gaithersburg, MD, USA), and nucleic acids passing through a UV recorder were detected at a wavelength of λ = 254 nm. Fractions of approximately 800 µL were collected, and RNAs were recovered from each fraction using phenol–chloroform–isoamyl alcohol (25:24:1, (*v*/*v*/*v*); Sigma-Aldrich, Taufkirchen, Germany, P2609), followed by ethanol precipitation.

### 4.2. miR-122 Overexpression and Inhibition in Huh-7 Cells

Huh-7 cells were transiently transfected with Lipofectamine RNAiMAX (Thermo Fisher Scientific, Meerbusch, Germany, 13778030) according to the manufacturer’s instructions with either 10 µM pre-miR-122 miRNA Precursor (mirVana miRNA mimic, Thermo Fisher Scientific, Meerbusch, Germany, 4464066), or 20 µM miR-122 antagomiR (Miravirsen, SPC-3649, Roche, Mannheim, Germany, formely Santaris Pharma) or scrambled oligo control (SPC-3744, Roche, Mannheim, Germany, formely Santaris Pharma). The sequence and efficacy of antagomiR (122-PM/SPC-3649) and scrambled oligo (SCR/ SPC-3744) were published by the authors in a previous study [3].

### 4.3. RNA Isolation, QC and qPCR Analysis

RNA isolation for qPCR analysis was performed with QIAzol Lysis Reagent (Qiagen, Hilden, Germany, 79306), followed by purification with miRNeasy Mini Kit (Qiagen, Hilden, Germany, 217004) according to the manufacturer’s instructions. RNA was quantified using a Qubit Fluorometer 2.0 (Thermo Fisher Scientific, Meerbusch, Germany). RNA isolation of PFA liver sections was performed as described in the Appendix A. cDNA synthesis for mRNA quantification was carried out as described in [3]. cDNA synthesis for miRNAs detection by qPCR was performed using the miQPCR method [23]. Notably, this method enables the universal reverse transcription of all the miRNAs contained in the reverse-transcribed RNAs. qPCR reactions were carried out on a StepOne Plus cycler (Thermo Fisher Scientific, Meerbusch, Germany), and amplicons were detected using SYBR Green I (GO-Taq PCR Master mix, Promega, Walldorf, Germany; A6002). Suitable reference genes were identified by using geNorm [55], and relative quantities were calculated using the ΔΔCt method [56]. qPCR data were analyzed by qBase v.1.3.5 [57]. Primers used for miRNA and mRNA quantification are listed in Appendix A.

### 4.4. Cloning and Target Validation by Dual-Luciferase Reporter Assay

*RNA22* [58] was used to predict potential miR-122-binding sites in the 3′UTRs of CEP55, CLIC1, EPS15L1, G6PD, KIF11, SLC1A5, and TK1 (Appendix A). Full-length 3′UTRs of human CEP55, CLIC1, G6PD, EPS15L1, KIF11, SLC1A5, and TK1 were amplified using an Expand High Fidelity PCR System (Roche, Mannheim, Germany, 11732641001) as indicated by the vendor. Primers used for 3′UTR amplification are listed in Appendix A. The resulting 3′UTRs were cloned into pMIR(+) and pMIR(−) reporter vectors as previously described [3]. Constructs carrying the 3′UTRs were transiently co-transfected with vectors carrying Renilla luciferase and either miR-122 mimic (122-MIM) or scramble oligos (SCR) into HEK293 cells with Lipofectamine RNAiMAX (Thermo Fisher Scientific, Meerbusch, Germany, 13778030) according to the manufacturer’s instructions. A dual-luciferase reporter assay (Promega, Walldorf, Germany, E1910) was conducted in accordance with the manufacturer’s instructions. The chemiluminescence assay was performed in white opaque 96-well plates with 50 µL of cell lysate and 50 µL of both LARII reagent (Firefly luciferase substrate) and Stop and Glo reagent (Renilla luciferase substrate), in a GloMax Multi Plus Multiplate Reader (Promega, Walldorf, Germany) according to the preset protocol with an integrity time of 10 s for each read. Data were normalized by calculating ratios of firefly/Renilla activities to correct for possible variations in transfection efficiencies.

### 4.5. Affymetrix Microarray and Gene Ontology Analyses

RNA was isolated from polyribosomal fractions as described above. In order to assess RNA integrity, 1 µL of individual polyribosomal RNAs were loaded on a PicoChip and run on a 2100 Bioanalyzer (Agilent Technologies, Santa Clara, CA, USA), according to the supplied protocol. Following QC and pooling, pooled RNAs were hybridized to Affymetrix Gene Chip Human Gene 1.0 ST arrays (Thermo Fisher Scientific, Meerbusch, Germany, 901086). cDNA syntheses, labeling, and hybridizations were performed by the genomics core facility of the European Molecular Biology Laboratory (EMBL, Heidelberg, Germany). Normalization and data analysis was carried out with AltAnalyze [15]. Microarray data are available via the Gene Expression Omnibus (GEO [59]) repository (accession number GSE234690). Gene ontology (GO) enrichment analysis was performed using ShinyGo [24] and GOrilla [27].

### 4.6. Protein Isolation, Western Blot Analysis, and Antibodies

Proteins were isolated using RIPA buffer with protease inhibitor cocktail (cOmplete Protease Inhibitor Cocktail, Roche, Mannheim, Germany, 04693116001), and concentrations were measured using the Qubit Protein Assay Kit (Thermo Fisher Scientific, Meerbusch, Germany, Q33211) according to the manufacturer’s instructions. Western blot analyses were carried out using 20 µg of total proteins. Protein lysates were loaded together with 10 µL of Protein Ladder (BioRad, Dusseldorf, Germany, 161-0373) on 10% or 12% SDS polyacrylamide gels and transferred to nitrocellulose membranes using semidry blotting systems according to standard protocols. Membranes were blocked with 5% milk powder (Carl Roth, Karlsruhe, Germany, T145.3) in Tris-buffered saline with Tween-20 (TBST) and incubated with incubation primary antibody for 1 h at RT or overnight at 4 °C. Subsequently, the appropriate horseradish peroxidase (HRP)-coupled secondary antibody was incubated for 2 h at room temperature. Chemiluminescence was detected with ECL Western blotting Substrate (Promega, Walldorf, Germany, W1001) using the ChemiDoc MP Imaging System (BioRad, Dusseldorf, Germany). Signal intensities of Western blot protein bands were analyzed using ImageLab (BioRad, Dusseldorf, Germany, version 6.0.1). Antibodies used in protein analysis by Western blot: anti-EPS15L1 antibody (Abcam, Cambridge, UK, ab53006), anti-G6PD antibody (Sigma-Aldrich Chemie GmbH, Taufkirchen, Germany, HPA000834), anti-β-actin antibody (Abcam, Cambridge, UK, ab8226), and anti-GAPDH antibody (AbD Serotec, Puchheim, Germany, MCA4739).

### 4.7. Sample Preparation and Proteome Analysis by Liquid Chromatography/Tandem Mass Spectrometry (LC–MS/MS)

Proteins were isolated from transfected cells or liver tissue as described above. For proteomics, LC–MS/MS was performed essentially as described by Schira et al. [60,61]. Detailed procedures and technical information are given in Appendix A.

### 4.8. Statistical Analysis and Imaging Software

Statistical analyses were carried out using GraphPad Prism (version 9.4.1). Data were expressed as the mean ± SD. Results compared between groups were analyzed using a two-tailed Student’s *t*-test when two samples were considered or using 1-way ANOVA for three or more samples. When groups were compared using 1-way ANOVA, we assumed that data were normally distributed. The data were considered significant at a *p*-value ≤ 0.05. Images were prepared using Affinity Designer (version 1.10.4.1198).

### 4.9. Data Mining from the Gene Expression Omnibus (GEO) Repository, and Data Analysis

The miRNA and mRNA profiling data from animal models and liver cancer patients were retrieved from the GEO repository [59]. Enrichment analysis was performed with ShinyGO [24] by uploading a list of significantly upregulated genes that were predicted to be miR-122 targets by *miRWalk* [22] target prediction database.

### 4.10. The Cancer Genome Atlas (TCGA) data and Kaplan–Meier Survival Curves

This study included data from TCGA Research Network (https://www.cancer.gov/tcga, accessed on the 2 February 2023), and used the KM Plotter [62] to generate Kaplan–Meier survival curves for hepatocellular carcinoma patients.

## 5. Conclusions 

In conclusion, our investigation of the role of miR-122 in liver function has revealed intriguing insights into its complex regulatory mechanisms. While the influence of miR-122 on liver processes has been widely recognized, its global impact on underlying gene networks and pathological pathways has remained elusive. Using novel approaches, we elucidated the potential of miR-122 in modulating key transcription factors such as NRF1 and E2F4, which are intrinsically linked to liver disease. Furthermore, our proteome analysis shed light on the involvement of miR-122 in promoting malignancy-associated protein changes. Specifically, this study establishes a connection between miR-122 and G6PD levels in hepatitis B virus-associated hepatocellular carcinoma, suggesting its clinical relevance. Taken together, these findings underscore the central role of miR-122 in maintaining hepatic balance and introduce a hitherto unexplored molecular network involving NRF1, E2F4 and YY1. We hypothesize that an interplay of signals orchestrates miR-122 dysregulation, potentially underlying inflammatory and cancerous conditions in the liver. In essence, our research brings a valuable piece to the intricate puzzle of liver homeostasis and disease progression, opening avenues for further exploration and therapeutic intervention.

## Figures and Tables

**Figure 1 cancers-15-04129-f001:**
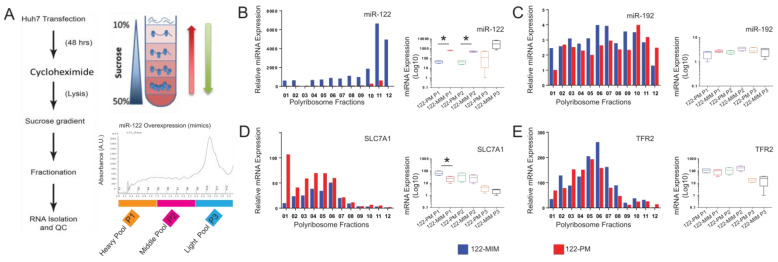
Direct identification of miR-122 targets by polyribosomal profiling on sucrose gradient. (**A**) Huh-7 cells were transfected with either miR-122 mimic (122-MIM) or miR-122 inhibitor (122-PM) and lysed 48 h after transfection. The cytosolic extracts were sedimented by ultracentrifugation on a 10–50% sucrose gradient. Upon fractionation, RNA isolation, quantification, and quality control (QC) were performed for the corresponding polyribosomal fractions. The UV absorbance profile of the nucleic acids associated with polyribosomes was measured at 254 nm from miR-122-overexpressing and miR-122-depleted cells. The collected fractions are depicted as (A1–A12). Analysis of miRNA and mRNA distribution in isolated polyribosomes of cells transfected with miR-122 mimics (blue) or cells treated with miR-122-antagomiR (red) by qPCR. (**B**) The analysis showed an increased level of miR-122 associated to polyribosomes in cells treated with miR-122 mimics (**C**), while the distribution of unrelated miRNAs as shown for miR-194 was unaffected by the transfection. (**D**) In polyribosomes isolated from antagomiRs treated cells, a significant enrichment toward the heavy transcribed fraction in the distribution of SLC7A1 mRNA, the best characterized target for miR-122, was observed. (**E**) In contrast, the transferrin receptor 2 (TFR2) mRNA, which is not an miR-122 target, did not show any significant changes in its polyribosomal distribution. qPCR data were median-normalized as described in [14]. Data are shown either as miRNA/mRNA expressions in individual fractions or as box plot using the mean ± SD for the pooled fractions (*n* = 4). Statistical analysis was performed with a two-tailed *t*-test; *p* ≤ 0.05 was considered significant. * *p* ≤ 0.05.

**Figure 2 cancers-15-04129-f002:**
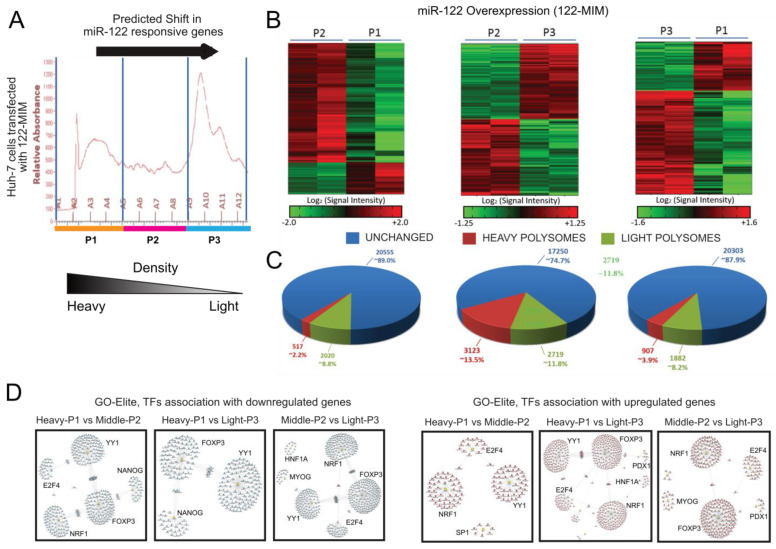
Changes in mRNA distribution on polyribosomes in response to miR-122 overexpression (122-MIM). Huh-7 cells were treated with miR-122 mimic and lysed 48 h later. Following sedimentation on 10–50% sucrose gradient, lysates were fractionated, and fractions were pooled as indicated (**A**) to create a heavy pool (P1), a middle pool (P2), and a light pool (P3). RNA isolated from each individual pool were hybridized to Affymetrix arrays, and data were analyzed by AltAnalyze [15] using a cutoff of 1.5 fold changes (significance level *p* ≤ 0.05, one-way ANOVA) The arrow indicates the expected target gene shift across polyribosomes in response to miR-122 overexpression. Microarray data are available via the GEO repository under the accession number GSE234690. (**B**) Hierarchical clustering representing the differential distribution of mRNAs associated with the polyribosomal pools (P2 vs. P1, P2 vs. P3, and P3 vs. P1). Cosine matrix was applied to generate a hierarchical tree of gene clusters (*n* = 2). (**C**) Cake chart illustrating the number of regulated transcripts between the different pools P2 vs. P1 (left), P2 vs. P3 (middle), and P3 vs. P1 (right). (**D**) Evaluation of microarray data by GO-Elite algorithm identified a link between miR-122-responsive genes and the nuclear respiratory factor 1 (NRF1), E2F transcription factor 4 (E2F4), transcription Forkhead box protein P3 (FOXP3), and Yin Yang 1 (YY1).

**Figure 3 cancers-15-04129-f003:**
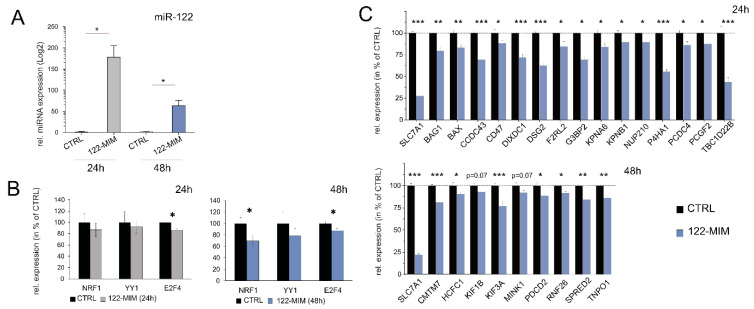
qPCR validation of potential miR-122 target genes in miR-122-overexpressing Huh-7 cells. (**A**) Human hepatoma cells Huh-7 were transfected with miR-122 mimic and RNA isolated 24 h and 48 h after transfection. The overexpression of cellular miR-122 was measured 24 h and 48 h after transfection by miQPCR [23], while miRNA expressions were normalized to miR-192. (**B**) Relative expression changes in the mRNA levels of NRF1, YY1, and E2F4 are depicted at 24 h and 48 h after miR-122 overexpression. (**C**) Relative expression changes for selected candidates are depicted at 24 h and 48 h post transfection. Death effector domain-containing protein (DEDD) mRNA was identified as the most stable transcript in the dataset using *GeNorm* algorithm for reference gene quality. The validated miR-122 target SLC7A1 mRNA served as a positive control. The values of control cells were set to 100%. Data are shown as the mean ± SD (*n* = 5). Statistical analyses were performed with a two-tailed *t*-test; *p* ≤ 0.05 was considered significant. ns = not significant; * *p* ≤ 0.05; ** *p* ≤ 0.01; *** *p* ≤ 0.001. Full gene names are listed in Appendix A.

**Figure 4 cancers-15-04129-f004:**
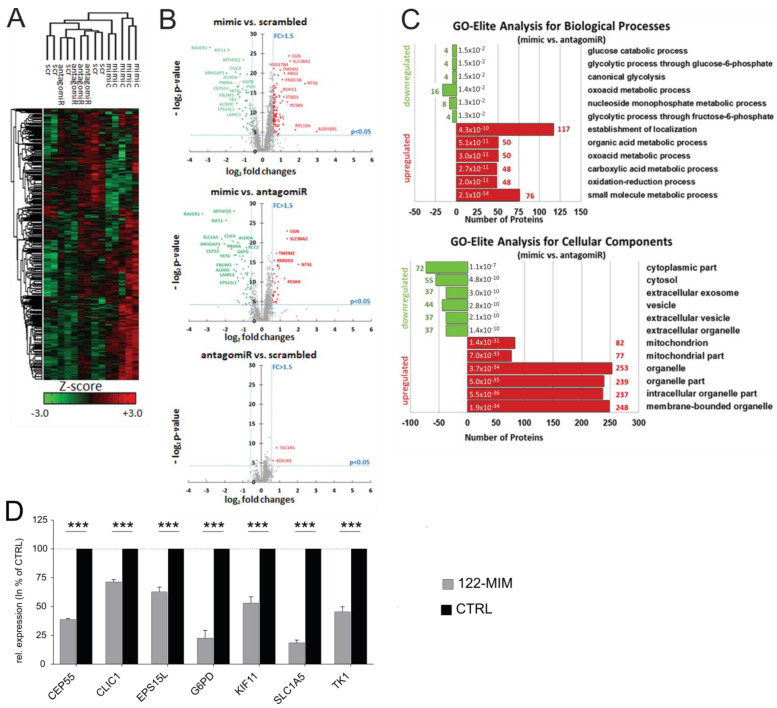
Proteome analysis of miR-122-enriched and -depleted Huh-7 cells. (**A**) Quantitative proteome analysis of Huh-7 cell lines transfected with miR-122 mimics, antagomiRs, or scrambled oligo control (scr). Unsupervised hierarchical clustering illustrating the relative abundance (Z-score) of all quantified proteins in Huh-7 upon modulation of cellular miR-122 levels (*n* = 5). (**B**) Volcano plot illustrating the pairwise differences in protein abundances in Huh-7 treated with miR-122 mimic-, antagomiR-, and scrambled oligo-transfected cells. Expression changes between groups are plotted in log_2_ scale on the abscissa (x-axis), while *p*-values measured by one-way ANOVA are plotted in log_2_ scale on the ordinate (y-axis). Thresholds for fold changes (FC ≥ 1.5 or FC ≤ −1.5) are shown as perpendicular dashed lines, while the threshold for statistical relevance (*p* ≤ 0.05) is shown as a horizontal dashed line. Proteins highlighted in green were found in significant lower abundance, while significantly more abundant proteins are highlighted in red. (**C**) Functional analysis of regulated proteins was conducted using Gene Ontology enrichment tool GOrilla [27]. The schematic representation illustrates the number of proteins associated with the six most significantly (i.e., with lowest *p*-value) depleted (green) and enriched (red) GO terms regarding biological process (upper) and cellular components (lower). The number of proteins related to the GO terms is given in red (enriched) or green (depleted) next to the bars, while the *p*-value for each GO term is written next to the y-axis. (**D**) Relative expression changes for selected target proteins at mRNA levels in response to miR-122 overexpression (122-MIM). Expressions were normalized to hypoxanthine phosphoribosyltransferase 1 (HPRT1). The values of scrambled control (CTRL) transfected cells were set to 100%. Data are shown as mean ± SD (*n* = 5). Statistical analyses were performed with a two-tailed *t*-test; *p* ≤ 0.05 was considered significant. *** *p* ≤ 0.001.

**Figure 5 cancers-15-04129-f005:**
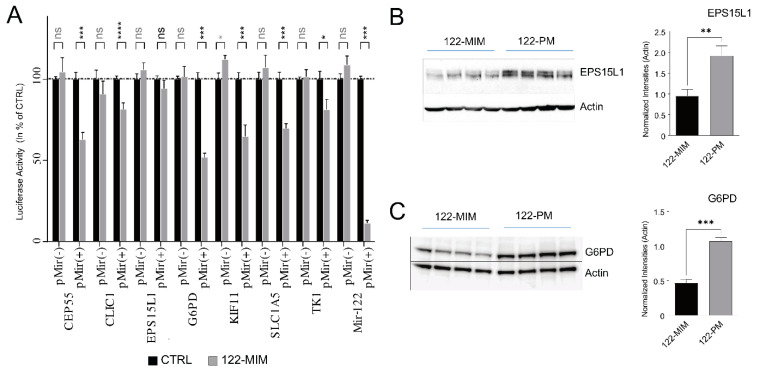
Correlation between miR-122 and G6PD expression in tumor tissue of HCC patients with or without HBV infection. (**A**) The full-length 3′UTRs for the seven GOIS were cloned into the luciferase promoter vector in the sense orientation [pMir(+)] or in the antisense orientation [pMir(−)] as a negative control. The plasmids pMir(+)_122 containing the perfect miR-122 binding site in sense orientation, and pMir(−)_122 (miR-122 binding site in the antisense orientation) served as the positive and negative controls, respectively. Plasmids were transfected in the presence (gray) or absence (black) of miR-122 mimic into HEK293 cells, and luciferase activity was measured 24 h post transfection. Relative luciferase activities are shown as a percentage of plasmid transfection control (*n* = 4). Western blot analysis demonstrated differential (**B**) EPS15L1 and (**C**) G6PD expression in Huh-7 treated with miR-122 mimic (122-MIM) or miR-122 inhibitor (122-PM) for 48 h (*n* = 4). Quantification of protein expression was normalized to actin. Statistical analyses were performed with a two-tailed *t*-test; *p* ≤ 0.05 was considered significant. ns = not significant; * *p* ≤ 0.05; ** *p* ≤ 0.01; *** *p* ≤ 0.001; **** *p* < 0.0001. The uncropped bolts are available in the Appendix A.

**Figure 6 cancers-15-04129-f006:**
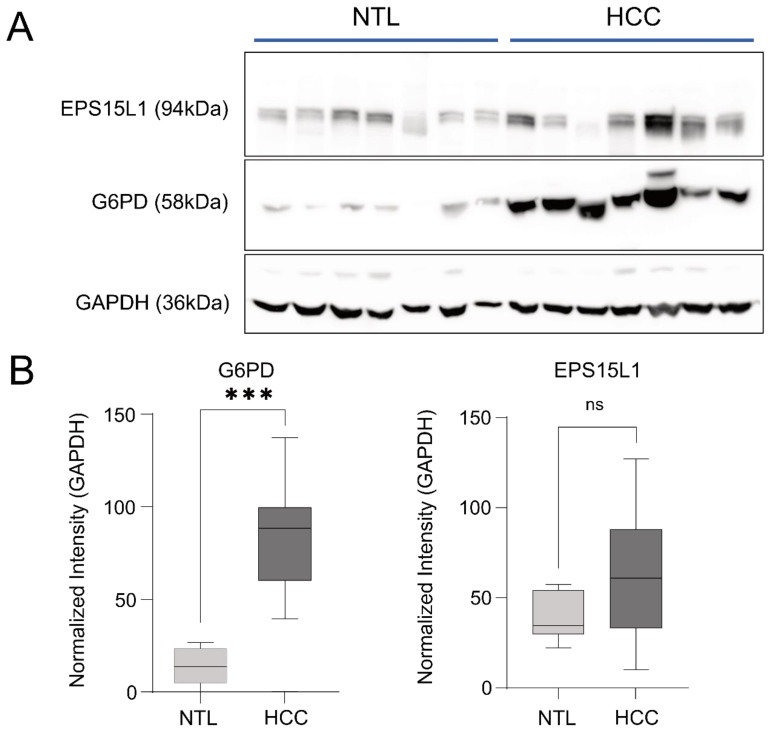
G6PD protein levels are significantly increased in the tumor tissue isolated from the livers of HCC patients. (**A**) Total proteins were isolated from the livers tissue of HCC patients, and the Western blot analysis for EPSL15L1 and G6PD was carried out in the tumor (HCC, *n* = 7) and in the non-tumor (NTL, *n* = 7) cells. (**B**) Quantification of protein expression was normalized to GAPDH. Statistical analyses were performed with s two-tailed *t*-test, *p* ≤ 0.05 was considered significant. ns = not significant; *** *p* ≤ 0.001. The uncropped bolts are available in Appendix A.

**Figure 7 cancers-15-04129-f007:**
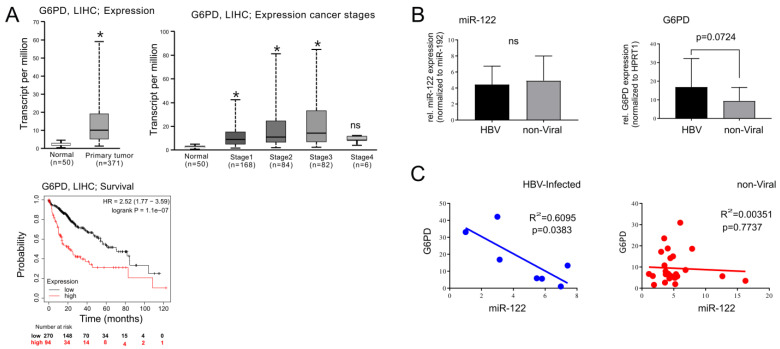
Correlation between miR-122 and G6PD expression in tumor tissue of HCC patients with or without HBV infection. (**A**) Analysis of LIHC cohorts from TCGA. Expression of G6PD was significantly increased in normal vs. primary tumor tissue (**Left panel**), as well as in the livers of patients with cancer stages 1 to 3 (**Right panel**). (**B**) Total RNAs were isolated from the FFPE liver tissue of HCC patients with (HBV-infected; *n* = 7) or without (non-viral: *n* = 26) HBV infection. Expression profiling of miR-122 (**Left panel**, normalized to miR-192) and G6PD (Right panel, normalized to HPRT1) were measured by using qPCR. (**C**) Linear regression analysis showed a significant (*p* = 0.0383) inverse correlation between miR-122 and G6PD in the livers of HBV-associated HCC (HBV-HCC) patients, whereas no correlation was found in non-viral HCC patients. A two-tailed *t*-test was used to compare two groups, One-way ANOVA was used to compare three or more groups, whereas linear regression (R^2^) was used to assess correlation between samples; *p* ≤ 0.05 was considered significant. ns = not significant; * *p* ≤ 0.05.

**Figure 8 cancers-15-04129-f008:**
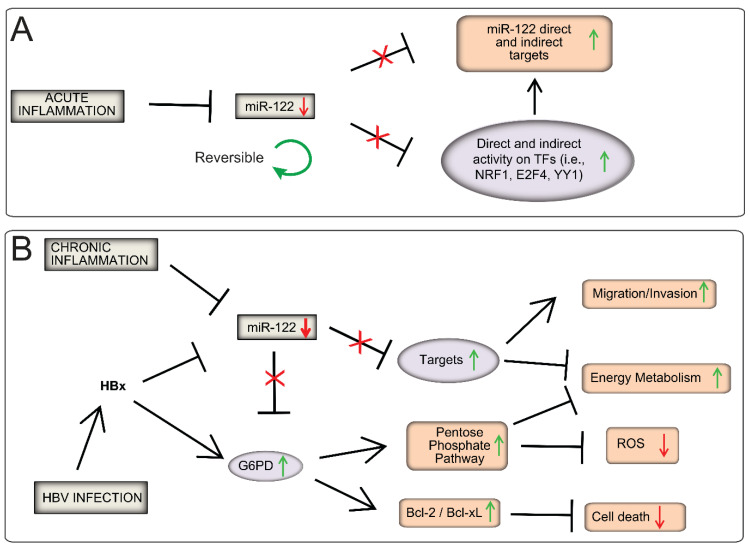
Proposed model for miR-122 in health and HBV-associated HCC. (**A**) Physiological regulation of miR-122: Physiological regulation of miR-122: miR-122 is transiently downregulated in response to (yet) unknown factors during inflammation, leading to upregulation of its targets and fine-tuning of gene expression downstream of NRF1, E2F4, and YY1 TFs. This model suggests that the transient downregulation of miR-122 leads to the fine-tuning of immune response and regeneration-associated networks. (**B**) Contribution of miR-122 in HBV-associated HCC: We propose that the dysregulation of miR-122 expression is a central factor in HCC development among HBV-infected patients. We propose that miR-122 transcription is inhibited by inflammation (**model A**). In addition, in HBV-infected patients, the viral-synthesized HBx has a dual role: (i) inhibiting miR-122 [41], and (ii) activating G6PD expression [33]. The combined downregulation of miR-122 and the activation of HBx leads to a “super activation” of G6PD, promoting cell survival (i.e., Bcl2/Bcl-xL activation [42]), energy metabolism [33], and ROS detoxification [43]. We propose that these changes might promote the transformation of hepatocytes to a malignant phenotype in HBV-infected patients.

## Data Availability

All study data are provided in the manuscript.

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
