# Peer review of "Uncovering Novel Roles of miR-122 in the Pathophysiology of the Liver: Potential Interaction with NRF1 and E2F4 Signaling"

_cancers, 2023, doi:10.3390/cancers15164129_

Round 1

Reviewer 1 Report

·      General comments: The research conducted by Paluschinski et al. makes an important contribution to the understanding of how miRNAs regulate pathophysiological pathways in relation to liver function. The nuclear respiratory factor-1 (NRF1) and the E2F Transcription Factor 4 (E2F4) are two specific transcription factors known to be dysregulated in liver disease. The authors used a novel method of transcriptome analysis of polyribosome bound RNAs to make this discovery. Additionally, by proteome analysis of hepatoma cells transfected with either a miR-122 mimic or antagomir 21, they uncovered changes in several proteins linked to an increase in malignancy that are known to be transcriptionally controlled by NRF1 and E2F4. This research sheds new light on potential molecular pathway changes that could manifest in the early stages of liver disease and are triggered by dysregulation.

However, I have few minor comments that authors should address in their revisions.

·         Specific comments: The following questions need to be answered to have a better understanding of this study and its potent future implications:

1.     Color codes in Fig 3C (page 6) do not indicate the data IDs.

2.     The experiments have been performed well in Huh-7 liver cancer cells, but it would be interesting to include an immortalized non-cancerous human hepatic cell lines such as HEK293 as a negative control.

3.     It is highly intriguing to learn that miRNA-122 regulates liver genes in a time dependent manner. It would be great to understand the underlying mechanism.

4.     Please highlight the control genes unaffected in normal cells vs hepatic cancer cells regulated by miR-122.

5.     Please improve the resolution of Figure 4B to make it more readable for the readers.

6.     Kindly emphasize the biological significance of the study (the invivo xenograft studies or 3-D culture models) in the text explaining the therapeutic potential of novel anti-miR-122 strategies against liver cirrhosis or HPV replication.

7.     Also, being an active cancer biologist working in the field of DNA damage and repair, I would like to learn more about the probable effects of miR-22 mediated regulation of NRF1 and E2F4, further influencing other physiological processes (DNA replication and repair) in liver cancer patients.

·         Overall English is fluent and easy to comprehend.

Author Response

Comments and Suggestions for Authors

  • General comments: The research conducted by Paluschinski et al. makes an important contribution to the understanding of how miRNAs regulate pathophysiological pathways in relation to liver function. The nuclear respiratory factor-1 (NRF1) and the E2F Transcription Factor 4 (E2F4) are two specific transcription factors known to be dysregulated in liver disease. The authors used a novel method of transcriptome analysis of polyribosome bound RNAs to make this discovery. Additionally, by proteome analysis of hepatoma cells transfected with either a miR-122 mimic or antagomir 21, they uncovered changes in several proteins linked to an increase in malignancy that are known to be transcriptionally controlled by NRF1 and E2F4. This research sheds new light on potential molecular pathway changes that could manifest in the early stages of liver disease and are triggered by dysregulation.

We sincerely appreciate the reviewer thoughtful comments and suggestions on our research manuscript. Your feedback reinforces our belief in the significance of our findings and their potential contribution to the field.

We will certainly take your suggestions into careful consideration as we revise and refine our manuscript. Your input will undoubtedly aid us in providing a more comprehensive and insightful analysis of the molecular pathways involved in early-stage liver disease. We are committed to addressing the points you've raised and are confident that your guidance will ultimately lead to a stronger and more robust publication.

Thank you once again for your time and expertise in reviewing our work. We look forward to incorporating your feedback and resubmitting an improved version of our manuscript.

However, I have few minor comments that authors should address in their revisions.

  • Specific comments: The following questions need to be answered to have a better understanding of this study and its potent future implications:

  1. Color codes in Fig 3C (page 6) do not indicate the data IDs.

  1. We express our gratitude to the reviewer for identifying the inaccuracy in the color-coding of Figure 3C. We have rectified this issue and updated the figure in the revised version of our manuscript.

  1. The experiments have been performed well in Huh-7 liver cancer cells, but it would be interesting to include an immortalized non-cancerous human hepatic cell lines such as HEK293 as a negative control.

  1. We appreciate the valuable feedback provided by the reviewer. We concur with the reviewer's observation that an investigation into the expression of miR-122 target genes in cell lines originating from non-hepatic sources, such as HEK-293 or HeLa, could potentially serve as a valuable negative control or offer insights into the roles of these genes in other tissue or tumor contexts.

Nonetheless, as rightly highlighted by the reviewer, the primary scope of our study centers on liver cancer. Given this specific focus, we thought that the inclusion of such experiments might inadvertently shift attention away from our main goals and potentially introduce confusion to the reader.

  1. It is highly intriguing to learn that miRNA-122 regulates liver genes in a time dependent manner. It would be great to understand the underlying mechanism.

  1. We share the reviewer's perspective that gaining a comprehensive understanding of the underlying mechanisms governing miR-122 regulation holds the promise of unveiling novel pathways suitable for targeted interventions. In alignment with this viewpoint, we are pleased to highlight our recent publication in Cells. In this study, titled “Differential Modulation of miR-122 Transcription by TGFβ1/BMP6: Implications for Nonresolving Inflammation and Hepatocarcinogenesis; https://doi.org/10.3390/cells12151955”, we delve into the intricate analysis of miR-122. Notably, our investigation reveals the distinct regulatory roles of Tgfb and Bmp6 in the modulation of miR-122 expression. Noteworthy is the observation that this regulatory mechanism remains conserved across diverse species, encompassing human, mouse, and rat.

  1. Please highlight the control genes unaffected in normal cells vs hepatic cancer cells regulated by miR-122.
  2. We would like to thank the reviewer for noticing that not all figures included the reference genes used to normalize the data. This error has now been corrected in the revised version of our manuscript.

  1. Please improve the resolution of Figure 4B to make it more readable for the readers.
  2. We are grateful to the reviewer for identifying the issue with the figures. In response, we have enhanced our manuscript by incorporating higher resolution images for all the figures.

  1. Kindly emphasize the biological significance of the study (the invivo xenograft studies or 3-D culture models) in the text explaining the therapeutic potential of novel anti-miR-122 strategies against liver cirrhosis or HPV replication.
  2. Based on the reviewer's feedback, we have expanded the discussion within the revised manuscript. This extended discussion, informed by our recent publication in Cells, thoroughly explores the prospects of inhibition of TGFβ signaling, elucidating its intricate involvement in the regulation of miR-122 and its associated target genes. In addition, as suggested by the reviewer, we also discussed the potential significance of our study in the context of the existing literature, including anti-miR-122 strategies to counteract liver cirrhosis and viral hepatitis.

  1. Also, being an active cancer biologist working in the field of DNA damage and repair, I would like to learn more about the probable effects of miR-22 mediated regulation of NRF1 and E2F4, further influencing other physiological processes (DNA replication and repair) in liver cancer patients.

  1. We express our appreciation for the reviewer's evident dedication to exploring cause-effects involving alterations in the molecular pathways that comprise progression to hepatocellular carcinoma. We too find our results stimulating and will certainly explore further the link between miR-122 and these transcription factors, as this may reveal new molecular mechanisms that could be therapeutic targets. However, as in our response to point 2, we decided not to further discuss the likely effects of miR-122-mediated regulation of NRF1 and E2F4, as this (in our opinion) might shift the focus away from our main message, which is centered on miR-122.

  • Overall English is fluent and easy to comprehend.

Reviewer 2 Report

In this manuscript, Paluschinski et al. investigate the functional activity of miR-122, the most abundant miRNA in hepatocytes, in pathophysiology of the liver. In general, the role of miRNAs in liver regeneration and injury has been shown previously, however key aspects of identification of novel targets and their therapeutic utility are just emerging.  Authors of current manuscript undertook two approaches to identify novel targets and elucidate their relevance - the polyribosome-bound RNA analysis in liver diseases and proteome analysis in hepatoma cells. Authors then identified CEP55, CLIC1, G6PD, K1F11, SLC1A5 and TK1 as novel targets of miR-122. They have further elaborated on the correlation of mR-122 and G6PD in HBV-infected HCC samples. On one hand, direct identification of mir-122 targets by polyribosomal profiling on sucrose gradient represent a novel aspects of the manuscript. Another strength of the manuscript is to their novel findings of miR-122 and its targets in regulation of energy metabolism and EV secretion. Overall, the manuscript presents reasonable findings that are relevant for liver cancer field, and the data support authors’ claims. Hence, I support the publication of these finding in their current format.

Author Response

Comments and Suggestions for Authors

In this manuscript, Paluschinski et al. investigate the functional activity of miR-122, the most abundant miRNA in hepatocytes, in pathophysiology of the liver. In general, the role of miRNAs in liver regeneration and injury has been shown previously, however key aspects of identification of novel targets and their therapeutic utility are just emerging.  Authors of current manuscript undertook two approaches to identify novel targets and elucidate their relevance - the polyribosome-bound RNA analysis in liver diseases and proteome analysis in hepatoma cells. Authors then identified CEP55, CLIC1, G6PD, K1F11, SLC1A5 and TK1 as novel targets of miR-122. They have further elaborated on the correlation of mR-122 and G6PD in HBV-infected HCC samples. On one hand, direct identification of mir-122 targets by polyribosomal profiling on sucrose gradient represent a novel aspects of the manuscript. Another strength of the manuscript is to their novel findings of miR-122 and its targets in regulation of energy metabolism and EV secretion. Overall, the manuscript presents reasonable findings that are relevant for liver cancer field, and the data support authors’ claims. Hence, I support the publication of these finding in their current format.

Answer to reviewer:

Dear Reviewer,

We extend our sincere gratitude to you, both on my own behalf and that of my co-authors, for dedicating your time to review our manuscript.

It brings us great satisfaction to learn that you have regarded our work with excellence and have deemed the manuscript suitable for publication in its current state.

Once more, we express our appreciation for your valuable contributions.
